# *Plasmodium falciparum* adapts its investment into replication *versus* transmission according to the host environment

**Abdirahman I Abdi[1,2]\*, Fiona Achcar[3,4], Lauriane Sollelis[3,4], João Luiz Silva-Filho[3,4], Kioko Mwikali[1], Michelle Muthui[1], Shaban Mwangi[1], Hannah W Kimingi[1], Benedict Orindi[1], Cheryl Andisi Kivisi[1,2], Manon Alkema[5], Amrita Chandrasekar[3], Peter C Bull[1], Philip Bejon[1], Katarzyna Modrzynska[3], Teun Bousema[5], Matthias Marti[3,4]\***

[1]KEMRI-Wellcome Trust Research Programme, Kilifi, Kenya; [2]Pwani University Biosciences Research Centre, Pwani University, Kilifi, Kenya; [3]Wellcome Center for Integrative Parasitology, University of Glasgow, Glasgow, United Kingdom; [4]Institute of Parasitology, Vetsuisse and Medical Faculty, University of Zurich, Zurich, Switzerland; [5]Radboud University Nijmegen Medical Centre, Nijmegen, Netherlands

**\*For correspondence:**
aabdi@kemri-wellcome.org
(AIA);
matthias.marti@glasgow.ac.uk
(MM)

**Abstract** The malaria parasite life cycle includes asexual replication in human blood, with a proportion of parasites differentiating to gametocytes required for transmission to mosquitoes. Commitment to differentiate into gametocytes, which is marked by activation of the parasite transcription factor *ap2-g*, is known to be influenced by host factors but a comprehensive model remains uncertain. Here, we analyze data from 828 children in Kilifi, Kenya with severe, uncomplicated, and asymptomatic malaria infection over 18 years of falling malaria transmission. We examine markers of host immunity and metabolism, and markers of parasite growth and transmission investment. We find that inflammatory responses associated with reduced plasma lysophosphatidylcholine levels are associated with markers of increased investment in parasite sexual reproduction (i.e. transmission investment) and reduced growth (i.e. asexual replication). This association becomes stronger with falling transmission and suggests that parasites can rapidly respond to the within-host environment, which in turn is subject to changing transmission.

## Editor's evaluation

This important work provides insight into how *Plasmodium falciparum* optimises the balance between infection of the human host and investment in onward transmission to the mosquito. The results, based on appropriate and validated state-of-the-art methodologies, are convincing and have both theoretical and practical implications beyond a single subfield.

## Introduction

Malaria remains one of the world's major public health problems. In 2021, an estimated 619,000 deaths and 247 million cases were reported (*WHO, 2022*). Around 70% of deaths are in African children under five years of age and are caused by a single parasite species, *Plasmodium falciparum* (*WHO, 2022*).

*P. falciparum* has a complex life cycle, involving obligatory transmission through a mosquito vector and asexual replication within erythrocytes of the human host. Between-host transmission requires the formation of gametocytes from asexual blood stage forms, as gametocytes are the only parasite stage to progress the cycle in the mosquito. A series of recent studies has demonstrated that commitment to gametocyte formation (i.e. stage conversion) is epigenetically regulated and occurs via activation of the transcription factor, AP2-G that in turn induces transcription of the first set of gametocyte genes (*Kafsack et al., 2014*; *Sinha et al., 2014*).

The parasites that do not convert into gametocytes continue to replicate asexually, contributing to within-host parasite population growth (i.e. parasite burden) and determining *P. falciparum* infection outcome that ranges from asymptomatic infections to severe complications and death (*Marsh and Snow, 1999*; *Langhorne et al., 2008*; *White, 2018*). Cytoadhesion of infected erythrocytes (IE) to receptors on microvascular endothelium of deep tissues reduces the rate of parasite elimination in the spleen (*Rowe et al., 2009*; *Turner et al., 2013*), thus supporting the within-host expansion of the parasite population (i.e. parasite burden). As a side effect of this parasite survival strategy, cytoadhesion reduces the diameter of the vascular lumen, thus impairing perfusion and contributing to severe malaria pathology (*Silamut et al., 1999*; *Taylor et al., 2004*; *Hanson et al., 2012*). *P. falciparum* erythrocyte membrane protein 1 (PfEMP1), encoded by the *var* multi-gene family, plays a critical role in both pathogenesis (through cytoadhesion) (*Smith et al., 1995*; *Su et al., 1995*) and establishment of chronic infection (through variant switching and immune evasion) (*Recker et al., 2004*; *Scherf et al., 2008*).

Both *var* gene transcription and stage conversion (and hence *ap2-g* transcription) are subject to within-host environmental pressures such as immunity (*Rono et al., 2018*), febrile temperature (*Oakley et al., 2007*; *Rawat et al., 2021*), nutritional stress (*Carter and Miller, 1979*), and drugs (*Buckling et al., 1999*), perhaps via a common epigenetic regulation mechanism (*Coleman et al., 2014*). For example, in vitro studies revealed that stage conversion can be induced by nutritional depletion such as spent culture media (*Carter and Miller, 1979*; *Williams, 1999*) and depletion of Lysophosphatidyl-choline (LPC) (*Brancucci et al., 2017*). Recent work from Kenya and Sudan provides some evidence that parasites in low relative to high transmission settings invest more in sexual commitment and less in replication and *vice versa* (*Rono et al., 2018*). Altogether these studies suggest that the parasite can sense and rapidly adapt to its environment in vitro and in vivo. A family of protein deacetylases called sirtuins is known to link environmental cues to various cellular processes via metabolic regulation (*Li, 2013*; *Bosch-Presegué and Vaquero, 2014*; *Vasquez et al., 2017*). They do this through epigenetic control of gene expression (*Vasquez et al., 2017*) and post-translational modification of protein function (*Vasquez et al., 2017*; *Zhu et al., 2012*). The *P. falciparum* genome contains two sirtuins (Pfsir2a/b) which have been linked to the control of *var* gene transcription (*Tonkin et al., 2009*; *Merrick et al., 2012*), and their expression is influenced by febrile temperature (*Oakley et al., 2007*) and low transmission intensity (*Rono et al., 2018*; *O'Meara et al., 2008*; *Mogeni et al., 2016*).

Here, we investigated the interplay between parasite and host environmental factors governing parasite investment in reproduction (to maximize between-host transmission) *versus* replication (to ensure within-host persistence) in vivo. We analyzed samples and clinical data collected from children in Kilifi county, Kenya, over changing malaria transmission intensity between 1994 and 2014. We quantified parasite transcripts for *ap2-g*, *PfSir2a*, and *var* genes, as well as *Pf*HRP2 protein levels (for parasite biomass) and levels of host inflammatory markers and lipid metabolites. We then integrated these host and parasite-derived parameters to interrogate their dynamics and interactions in the context of changing transmission intensity and immunity.

## Results

### A clinical malaria patient cohort across changing transmission periods in Kilifi, Kenya

The study included samples and clinical data collected from 828 children from Kilifi county, Kenya, over 18 years of changing malaria transmission (*O'Meara et al., 2008*; *Mogeni et al., 2016*). The study period encompassed three defined transmission phases: pre-decline (1990–2002), decline (2003–2008), and post-decline (2009–2014) (*Figure 1A*). During the study period, a total of 26,564 malaria admissions were recorded at Kilifi county hospital (*Figure 1A and B*). While the number

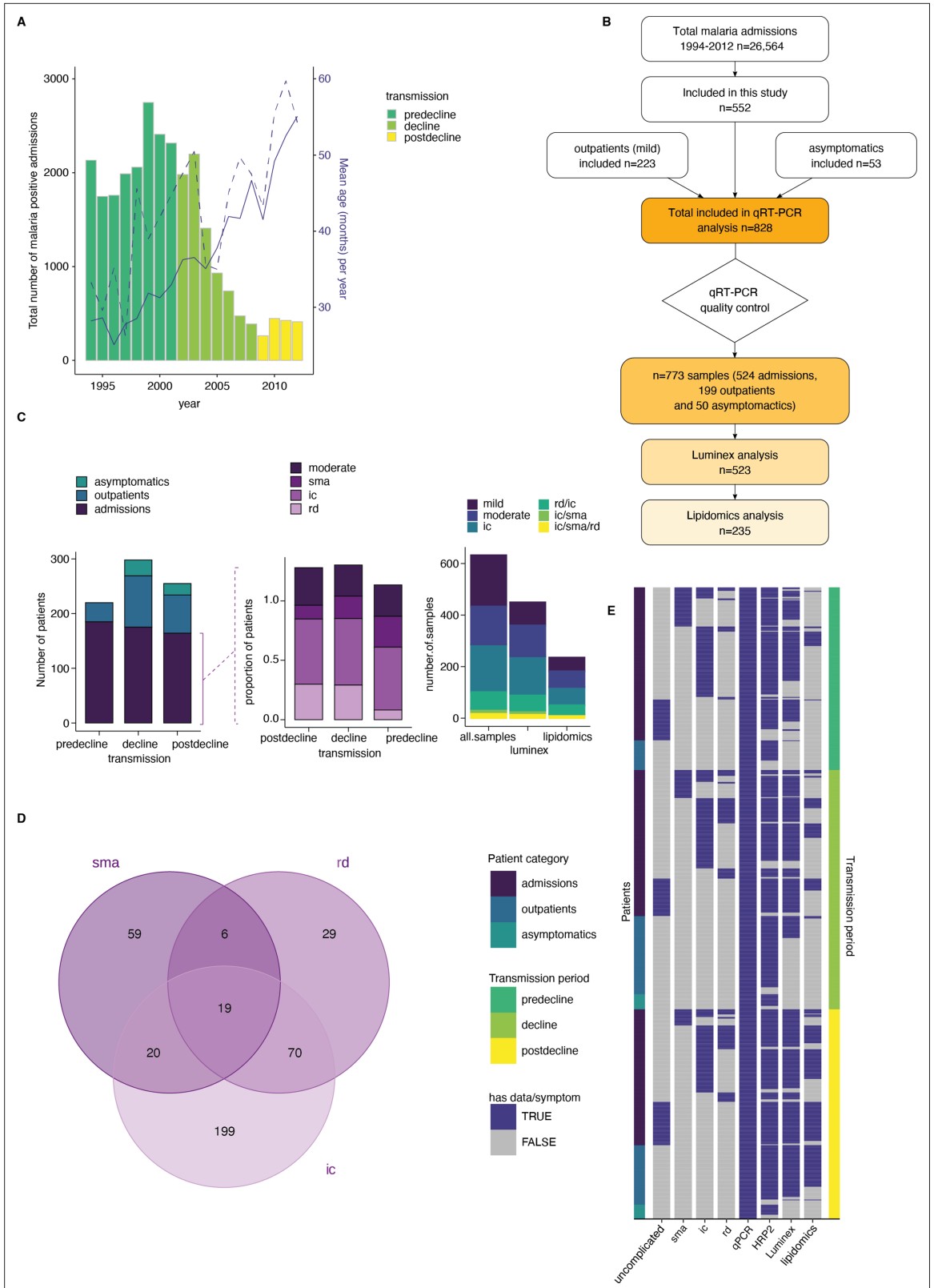

**Figure 1.** A clinical malaria patient cohort during changing the transmission in Kilifi, Kenya. (**A**) Total malaria admissions and patient age of the parent cohort. Number of patients per year (gray histogram, left axis). The solid blue line is the average patient age in the parent cohort, the dashed line is the average patient age in this study (both right axis). (**B**) Schematic of sample selection for this study. (**C**) Clinical presentation of patients selected for this study. Left: all patients, middle: admissions only, right: subset selected for luminex and lipidomics analysis. sma = severe malarial anemia, ic = impaired

*Figure 1 continued on next page*

*Figure 1 continued*

consiousness, rd = respiratory distress. (**D**) Number of patients in this study with different clinical presentations (402 severe cases initially selected). (**E**) Overview of the data available for each patient of the study, after excluding samples with *Pfsir2a* and *ap2-g* transcript transcription units greater or equal to 32 as described in the methods. Each row is one patient, organized by patient category (left axis) and transmission period (right axis).

of parasite-positive admissions decreased, the mean patient age at admission increased over time (*Marsh and Snow, 1999*; *White, 2018*; *Njuguna et al., 2019*; *Figure 1A*). For our study, 552 of the admissions were pragmatically selected to ensure adequate sampling of the transmission periods and clinical phenotypes (*Figure 1C*). 150 patients presented with moderate malaria and 402 with one or a combination of the severe malaria syndromes: impaired consciousness (IC), respiratory distress (RD), and severe malaria anemia (SMA) (*Marsh et al., 1995*; *Figure 1D*). 223 samples from children presenting with mild malaria at outpatient clinics and 53 asymptomatic children from a longitudinal malaria cohort study were added to cover the full range of the possible outcomes of malaria infection (*Figure 1B–E*), bringing the total number included in this study to 828 children. The characteristics of participants and clinical parameters are summarized in *Supplementary file 1*.

## Dynamics of parasite parameters across transmission period and clinical phenotype

First, we analyzed the dynamics of parasite parameters across transmission periods and clinical outcomes. For this purpose, we measured both total parasite biomass based on *Pf*HRP2 levels (*Dondorp et al., 2005*) and peripheral parasitemia based on parasite counts from blood smears. Total parasite biomass (*Pf*HRP2) but not peripheral parasitemia decreased with declining transmission (*Figure 2A*). This decrease was significant in the patients presenting with mild malaria at outpatient clinics which is a more homogenous clinical subgroup as compared to admissions consisting of a range of clinical phenotypes (*Figure 1C–D*).

Parasite samples were subjected to qRT-PCR analysis to quantify transcription of *ap2-g*, *Pfsir2a*, and *var* gene subgroups relative to two housekeeping genes (fructose biphosphate aldolase and seryl tRNA synthetase) (*Salanti et al., 2003*; *Rottmann et al., 2006*; *Lavstsen et al., 2012*; *Abdi et al., 2016*). In line with recent findings (*Rono et al., 2018*), *ap2-g* transcription increased significantly with declining malaria transmission (*Figure 2B*, **left**). In the subsequent analysis, only clinical cases were considered. Asymptomatic patients were excluded except for analysis of clinical phenotype since asymptomatic sampling was limited to the decline and post-decline periods (*Figure 1C*), and, therefore, data could not be corrected for transmission.

Further analysis demonstrated that *ap2-g* transcription is highly significantly correlated with transcription levels of the gametocyte marker *Pfs16* (*Figure 2C*). This association validates *ap2-g* as a proxy for both, stage conversion and gametocyte levels. *Pfsir2a* transcription followed the same trend across transmission periods (*Figure 2B*, **right**) and was positively associated with *ap2-g* transcription (*Figure 2D*). *Pfsir2a* and *ap2-g* transcription also showed a positive association with fever (*Figure 2E*), suggesting that both factors are sensitive to changes in the host inflammatory response. *Pfsir2a* but not *ap2-g* transcription also showed a significant negative association with *Pf*HRP2 (*Figure 2D*). Given this unexpected observation, we investigated the well-established associations between *Pfsir2a* transcription and *var* gene transcription patterns (*Merrick et al., 2012*; *Abdi et al., 2016*). *Pfsir2a* transcription showed a positive association with global upregulation of *var* gene transcription, particularly with subgroup B (*Figure 2F*). Likewise, transcription of group B *var* subgroup, *Pfsir2a*, and *ap2-g* transcription followed a similar pattern in relation to clinical phenotypes (*Figure 2—figure supplement 1*). To ensure that there is no systematic bias in the qRT-PCR data, we also tested the association of *Pfs16* expression against *var* gene subgroups. *Pfs16* expression showed a significant positive correlation with group B (here termed gpb1) and C (gpc2) *var* genes, while no correlation was observed with the rest of the *var* subgroups (*Figure 3—figure supplement 1A*). Altogether these data suggest co-regulation of *ap2-g* and *Pfsir2a* and a negative association between *Pfsir2a* and *Pf*HRP2, likely through host factors that are changing with the declining transmission.

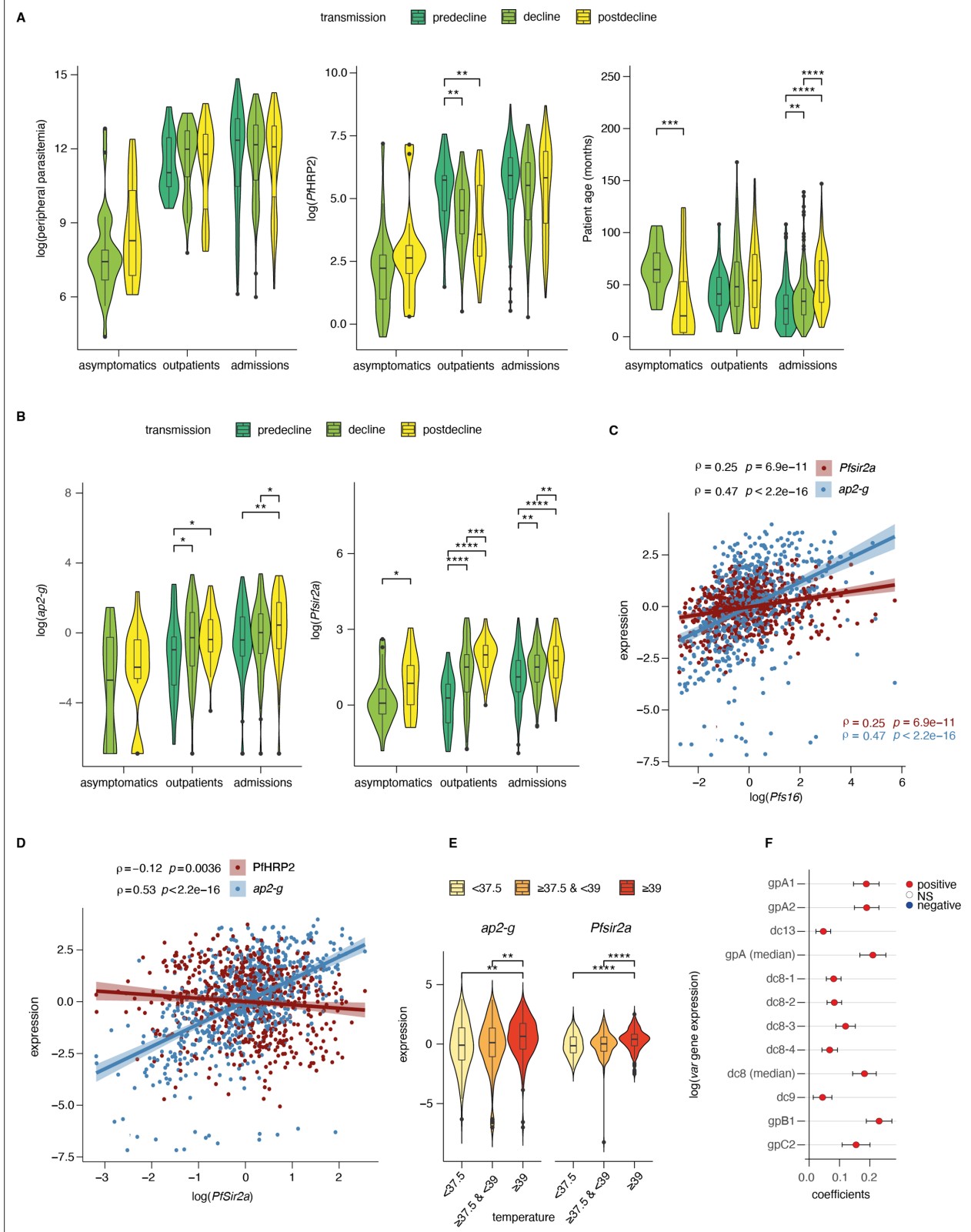

**Figure 2.** Dynamics of parasite parameters across transmission periods. (**A**) Peripheral parasitemia (smear, left), total parasite biomass (*Pf*HRP2, middle), and patient age (right) across patients. Number of patients: Asymptomatics: decline: n=29, postdecline: n=21; Outpatients: predecline: n=35, decline: n=94, postdecline: n=70; Admissions: predecline: n=185, decline: n=175, postdecline: n=164. (**B**) *ap2-g* transcript levels (left) and *Pfsir2a* levels (right) across patients. (**C**) Spearman's correlation between *Pfs16* and *ap2-g* (blue), or *PfSir2a* transcription (red) across patients (corrected for transmission). The

*Figure 2 continued*

lines fitted are linear regressions for visualization only. (**D**) Spearman's correlation between *Pfsir2a* and *ap2-g* transcription (blue), or *Pf*HRP2 levels (red) across patients (corrected for transmission). The lines fitted are linear regressions for visualization only. (**E**) *ap2-g* and *Pfsir2a* transcription (corrected for transmission) stratified by patient temperature. "<37.5": n=122, ">=37.5 & <39": n=228, ">=39": n=196. (**F**) Linear regression showing the association of *var* gene transcription levels with *Pfsir2a* levels (adjusted for transmission). 95% confidence intervals are shown. n=723 for all but c2 n=577. The color indicates whether the relationship is statistically significant (with Benjamini & Hochberg multiple tests correction). Positive correlations are in red, and negative in blue. gpA1, gpA2, and dc13 represent group A *var* gene transcripts. dc8-1 to dc8-4 represent DC8 *var* gene transcripts while gpb1 and gpc2 represent group B and C *var* genes. In the above graphs C-F (and all subsequent figures), asymptomatics were excluded in analyses involving the transmission period since they are not represented in the pre-decline period. All pairwise statistical tests indicated in the graphs are Wilcoxon tests corrected for multiple testing (Benjamini & Hochberg, *=FDR < 0.05, **=<0.01, ***=<0.001, and ****=<0.0001).

The online version of this article includes the following figure supplement(s) for figure 2:

**Figure supplement 1.** Parasite parameters stratified by clinical categories.

## *ap2-g* and *Pfsir2a* transcription is associated with a distinct host inflammation profile

We hypothesized that the observed variation in *ap2-g* and *Pfsir2a* levels across the transmission period and clinical phenotype is due to underlying differences in the host inflammatory response. To test this hypothesis, we quantified 34 inflammatory markers (*Huang et al., 2017*) with Luminex xMAP technology in the plasma of the 523 patients from the outpatient and admissions groups (*Figure 1B*). These patients were selected from the original set of 828 to ensure adequate representation of the transmission periods and clinical phenotypes (including fever), as summarized in *Figure 1C and E*. For this analysis, all associations were corrected for patient age and *Pf*HRP2 levels as possible confounders.

The markers MCP-1, IL-10, IL-6, and IL-1ra were significantly positively correlated with *ap2-g* and *Pfsir2a* transcription (*Figure 3A* and *Figure 3—figure supplement 1B*). To cluster the inflammatory markers based on their correlation within the dataset, we used exploratory factor analysis and retained five factors with eigenvalues above 1 (*Figure 3—figure supplement 2*). Factor loadings structured the inflammation markers into five profiles with distinct inflammatory states (*Figure 3B*). F1 consists of a mixture of inflammatory markers that support effector Th1/Th2/Th9/Th17 responses (i.e. hyperinflammatory state), F2 represents a Th2 response, F3 represents markers that support follicular helper T cell development and Th17 (*Dong, 2021*; *Chao et al., 2023*), F4 represents markers of immune paralysis/tissue-injury linked to response to cellular/tissue injury (*Kumar et al., 2014*), and F5 represents the inflammasome/Th1 response (*Weiss et al., 2018*). F4 showed a significant positive association with *ap2-g* and *Pfsir2a* transcription and fever (*Figure 3C*). In contrast, F5 showed a negative association with *ap2-g* and fever while F1 was positively associated with fever (*Figure 3C*). In parallel with the observed decrease in *Pf*HRP2 levels (*Figure 2A*), F1 and F5 significantly declined with falling transmission (*Figure 3D*).

The data support our hypothesis and suggest that the host inflammatory response changes with the falling transmission. Of note, the observed negative association between *Pfsir2a* transcription and *Pf*HRP2 levels appears to be independent of the measured cytokine levels (*Figure 3—figure supplement 3*) and is hence likely the result of parasite intrinsic regulation of replication.

## Plasma phospholipids link variation in the host inflammatory profile to *ap2-g* and *Pfsir2a* transcription

We have previously demonstrated in vitro that the serum phospholipid LPC serves as a substrate for parasite membrane biosynthesis during asexual replication, and as an environmental factor sensed by the parasite that triggers stage conversion (*Brancucci et al., 2017*). Plasma LPC is mainly derived from the turnover of phosphatidylcholine (PC) via phospholipase A2, while in the presence of Acyl-CoA the enzyme LPC acyltransferase (LPCAT) can drive the reaction in the other direction (*Law et al., 2019*; *Amunugama et al., 2021*). LPC is an inflammatory mediator that boosts type 1 immune response to eliminate pathogens (*Huang et al., 1999*; *Qin et al., 2014*). LPC turnover to PC can be triggered by inflammatory responses aimed to repair and restore tissue homeostasis rather than eliminate infection (*Law et al., 2019*). Here, we performed an unbiased lipidomics analysis of plasma from a representative subset of the outpatient and admission patients (*Figure 1* and *Figure 4—figure supplement 1*)

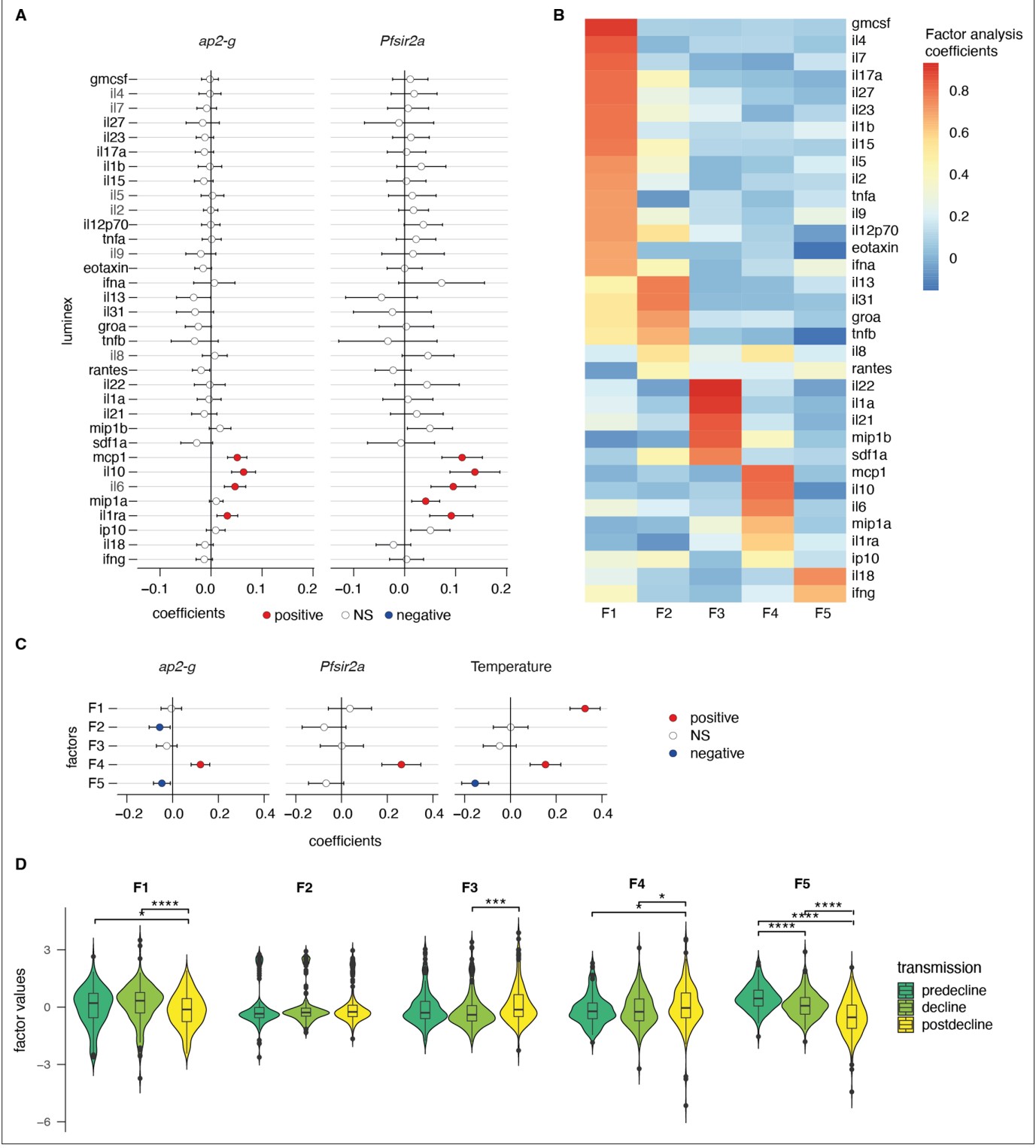

**Figure 3.** *ap2-g* and *Pfsir2a* transcription levels are associated with the host inflammation profile. (**A**) Association of inflammatory markers with *ap2-g* and *Pfsir2a* transcripts, tested using transmission period, age, and *Pf*HRP2 adjusted linear regression (*p*-values adjusted for multiple testing using Benjamini & Hochberg multiple tests correction). Plotted is the regression coefficient (estimate) and 95% CI. Above and below zero indicate statistically significant positive (red) and negative associations (blue), respectively. n=523 (**B**) Principal exploratory factor analysis. The figure shows the inflammatory marker loadings on the five factors (**F1–F5**) identified to have eigenvalue above 1. (**C**) Linear regression between inflammatory factors. (**F1–F5**) and *ap2-g* and *Pfsir2a* transcription and patient temperature (adjusted for transmission, *Pf*HRP2, and age). Plotted is the coefficient between the factor and the parameter (estimate) and 95% CI. Number of Patients: *ap2-g* and *Pfsir2a*: n=523, Temperature: n=496. The association is significant if the correlation

*Figure 3 continued on next page*

*Figure 3 continued*

FDR <0.05, in which case the positive associations are marked in red and the negative ones in blue. (**D**) Inflammatory factors stratified by transmission period. Pairwise tests are Wilcoxon tests (Benjamini & Hochberg, *=FDR < 0.05, **=<0.01, ***=<0.001, and ****=<0.0001). Number of patients: predecline: n=131, decline: n=180, postdecline: n=212.

The online version of this article includes the following figure supplement(s) for figure 3:

**Figure supplement 1.** Parasite and inflammatory markers stratified by transmission period.

**Figure supplement 2.** Factor loadings.

**Figure supplement 3.** Correcting *Plasmodium falciparum* histidine-rich protein 2 (*Pf*HRP2) vs *Pfsir2a* associations for external factors.

to explore whether the host inflammatory profile modifies the plasma lipid profile and consequently *ap2-g* and *Pfsir2a* transcription levels in vivo.

We examined associations between the host inflammatory factors (F1-F5) and the plasma lipidome data. Again, these associations were corrected for transmission period, patient age, and *Pf*HRP2 levels. 24 lipid species dominated by phospholipids, showed significant association with the inflammatory factors at a false discovery rate below 0.05 (*Figure 4A*). Like the observed associations with *ap2-g* and *Pfsir2a* transcription, cytokines in the F4 and F5 factors showed reciprocal associations with various LPC species and PC (*Figure 4A*): F5 showed a significant positive association with one LPC species and negative associations with PC, respectively, while F4 showed a significant negative association with three LPC species (*Figure 4A*, *Supplementary file 2*). The positive association of LPC with the F5 inflammatory factor is consistent with previous findings that identified LPC as an immunomodulator that can enhance IFN-γ production and the activation of the inflammasome (*Huang et al., 1999*; *Qin et al., 2014*), which results in increased levels of cytokines such as IL-18 (*Weiss et al., 2018*) and is necessary for eliminating parasites. Depletion of LPC is also associated with elevated markers of tissue injury (F4), perhaps following uncontrolled parasite growth or maladaptive inflammation. In summary, the association of inflammatory factors with lipids identified LPC, PC, and PE species as the most significant ones (*Figure 4A*), in line with their known immunomodulatory role. Importantly, we observed the same pattern in a controlled human infection model where parasite densities were allowed to rise to microscopic levels, both after sporozoite, and blood-stage infection (*Figure 4B* and *Figure 4—figure supplement 2*; *Alkema et al., 2021*; *Alkema et al., 2022*). Next, we examined the main lipid species associated with the five inflammatory factors with respect to *ap2-g* and *Pfsir2a* transcription. Indeed, LPC species showed a negative association with both *ap2-g* and *Pfsir2a* transcription levels (*Figure 4C–F*). The association was only significant in our data when inflammation is highest (and LPC level lowest), which is at low transmission (i.e. post decline). In contrast, LPC levels are highest and vary little across patients at high transmission (pre-decline).

Altogether, these data provide in vivo evidence for the previously observed link between LPC depletion and *ap2-g* activation and strongly suggest that LPC is both, a key immune modulator and a metabolite whose level is sensed by the parasite. Importantly, the key relationships described in *Figures 2–4* were independently significant in a structural equation model that examined how host immunity modifies the host-parasite interaction, the within-host environment, and parasite investment in transmission or replication (*Supplementary file 3*).

## Discussion

Malaria parasites must adapt to changing environmental conditions across the life cycle in the mammalian and mosquito hosts. Similarly, changing conditions across seasons and transmission settings require both within- and between-host adaptation to optimize survival in the human host *versus* transmission to the next host. First, a recent transcriptomic study from Kenya and Sudan suggested that parasites in low transmission settings (where within-host competition is low) invest more in gametocyte production compared to high transmission settings (where within-host competition is high) (*Rono et al., 2018*). Second, a longitudinal study from Senegal demonstrated that human-to-mosquito transmission efficiency (and gametocyte density) increases when parasite prevalence in the human population decreases, suggesting that parasites can adapt to changes in the environment (*Churcher et al.,*

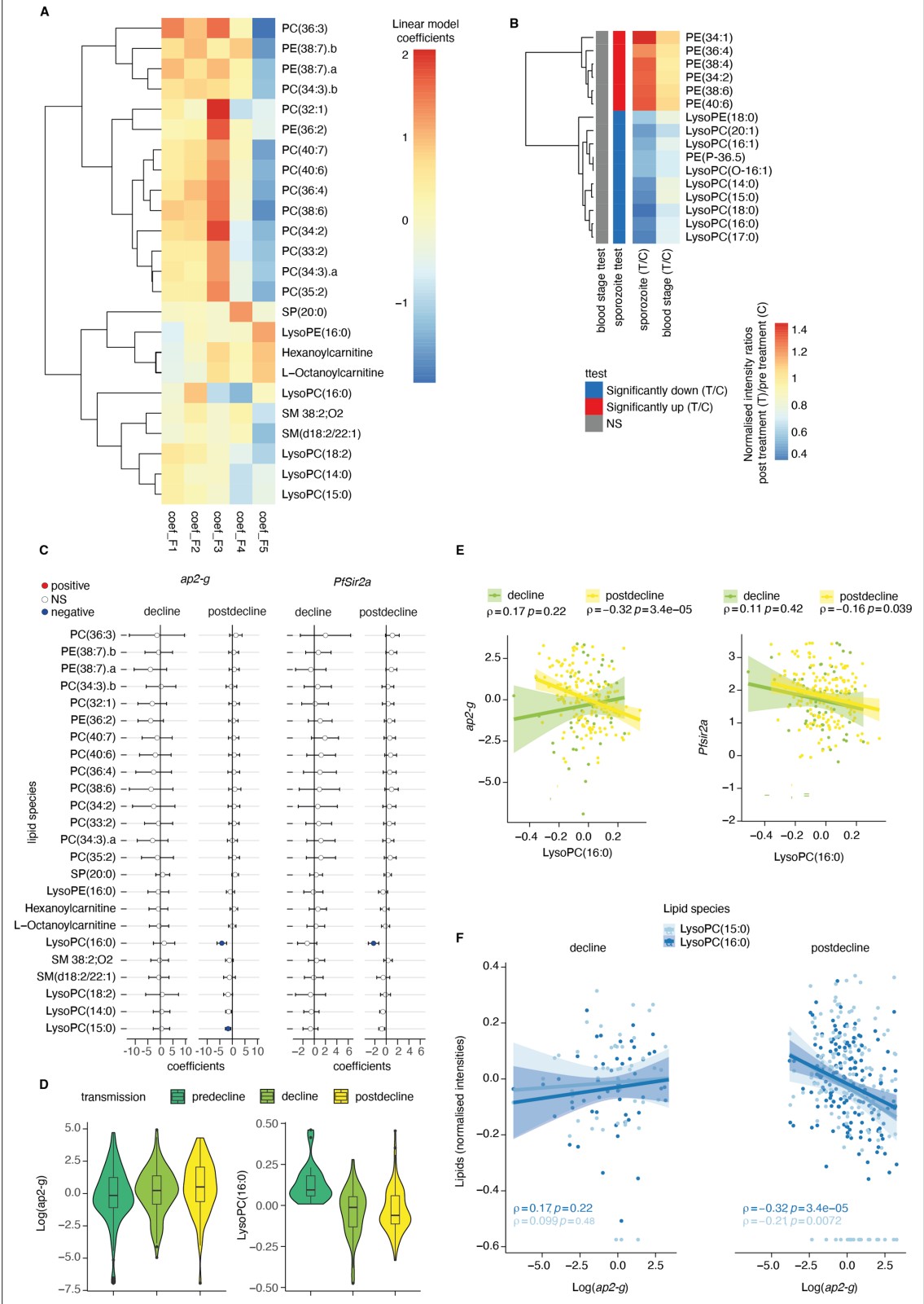

**Figure 4.** Plasma Lysophosphatidylcholine (LPC) links host inflammation to *ap2-g* and *Pfsir2a* transcription. (**A**) Heatmap of the linear regression coefficients between lipids and inflammatory factors (F1-F5, adjusted for transmission period and corrected for multiple testing). Shown are all lipids that are significantly associated (positive or negative) with factors F1-F5, clustered using R hclust (distance = Euclidean, method = centroid) and that have been manually identified and filtered for peak quality (isotopes and fragments weres also filtered out). (**B**) Shown are the lipids with significant

*Figure 4 continued on next page*

*Figure 4 continued*

differences (student's t-test corrected for multiple testing) between pre- and post-treatment in the controlled human malaria infections (CHMI) for either infection route (blood or sporozoite infection). Plotted is the fold-change post-treatment vs pre-treatment. On the left is indicated whether the lipid is significantly increased (red) or decreased (blue) in either route of infection. (**C**) Linear regression between the lipids from A and *ap2-g* or *Pfsir2a* transcription levels. Plotted is the coefficient and 95% CI. Blue indicates statistically significant negative correlations, while red indicates statically significant positive correlations (FDR <0.05). (**D**) Distribution of *ap2-g* transcript levels (left) and LysoPC (16:0) intensity (right) across patients and stratified by transmission period. Data are corrected for age and *Pf*HRP2. (**E**) Correlation between LPC (16:0) (top) and *ap2-g* (top), or *Pfsir2a* (bottom) transcription (Spearman's correlations corrected for multiple testing). (**F**) Correlations between identified LPCs and *ap2-g* transcription by transmission period (Spearman's correlations corrected for multiple testing). Note that the predecline period is not plotted separately in panels C-E due to insufficient sample numbers for the statistical analysis. Number of patients: predecline: n=22, decline: n=53, postdecline: n=160.

The online version of this article includes the following figure supplement(s) for figure 4:

**Figure supplement 1.** Sample subsetting and batch correction for lipidomics data.

**Figure supplement 2.** Controlled human malaria infections (CHMI) data.

*2015*). However, the within-host mechanisms driving parasite adaptation to the prevailing environment remain unclear.

Here, we analyzed parasite and host signatures in the plasma from a large malaria patient cohort over 18 years of declining malaria transmission in Kenya. This investigation allowed us to define some of the within-host environmental factors that change with transmission intensity and consequently influence the parasite's decision to invest in reproduction *versus* replication. A major strength of our study is that observations are from a single site and are thus plausibly reflective of transmission-related changes in parasite investments, rather than differences between geographically distinct parasite populations. We show that high transmission is associated with a host immune response that promotes parasite killing without compromising the intrinsic replicative ability of the individual parasite. In contrast, low transmission is associated with a host immune response that increases within-host stressors (fever, nutrient depletion), which trigger higher parasite investment into transmission (see also the model in *Figure 5*). Importantly, the observed associations between the parasite parameters *ap2-g*, *Pfsir2a*, and host inflammation remain significant if corrected for transmission, but they are strongest at low transmission (i.e. post decline period) when inflammation and the risk of damaging the host are highest. We anticipate that the proposed model could be tested in controlled human

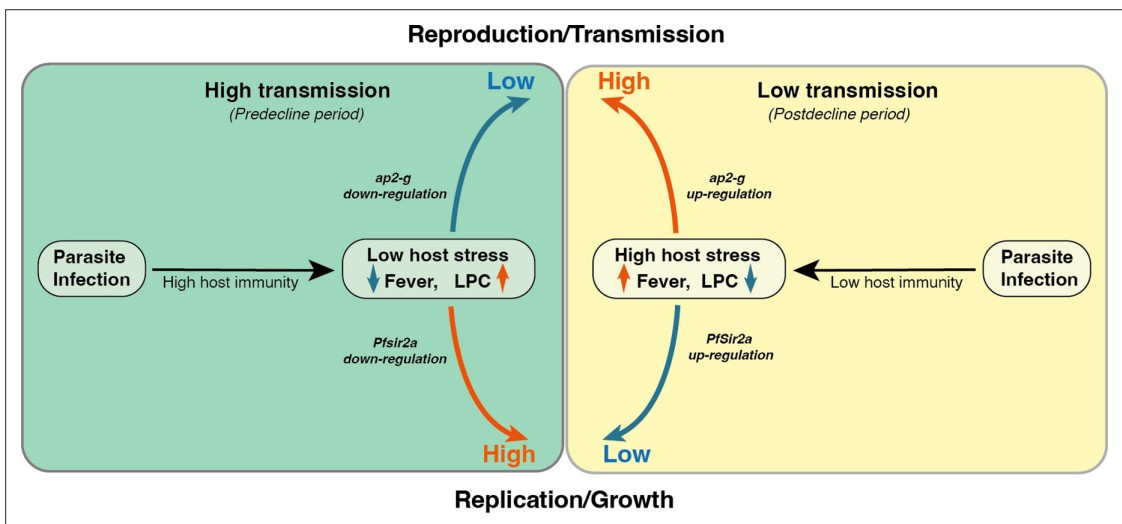

**Figure 5.** Proposed model on a within-host adaptation of the parasite to changing environments. The model is based on the interaction between the different host and parasite parameters described in this study. It proposes that declining transmission reduces host immunity, resulting in an inflammatory response associated with increased host stressors (including reduced Lysophosphatidylcholine (LPC) availability, and fever) and susceptibility to clinical symptoms/damage. The altered host response modifies the parasite response during infection, resulting in increased investment in transmission (as indicated by the elevated *ap2-g* levels) and reduced replication (as indicated by elevated *Pfsir2a* levels and reduced parasite burden/*Plasmodium falciparum* histidine-rich protein 2 (*Pf*HRP2) levels).

malaria infections with malaria naïve and semi-immune participants as a proxy for low vs high transmission settings.

At a systemic level, inflammation can influence the within-host environment and modulate parasite investment in replication *versus* reproduction by altering the levels of environmental stressors (e.g. oxidative, thermal, or nutritional stress). Consistent with this hypothesis, we show that a pro-inflammatory response mediated by IFN-γ/IL-18 (F5 in our analysis) promoting pathogen killing (*Weiss et al., 2018*; *Hoffman et al., 1997*; *Kearney et al., 2013*) is negatively associated with *ap2-g* and *Pfsir2a* transcription. In contrast, inflammatory markers that increase within-host environmental stress (e.g. fever) or reflect the extent of host tissue injury and are secreted to heal and restore homeostasis rather than kill pathogens (F4) (*Shapouri-Moghaddam et al., 2018*; *Tan et al., 2021*) are positively associated with *ap2-g* and *Pfsir2a* transcription. At a metabolic level, we previously demonstrated that LPC depletion induces *ap2-g* transcription and, therefore, gametocyte production in vitro (*Brancucci et al., 2017*). A recent study has provided the first indications of a possible association between LPC and *ap2-g* levels in a small malaria patient cohort (*Usui et al., 2019*). Here, we reveal that LPC levels are negatively associated with *ap2-g* transcription in patient plasma, thus providing direct evidence for our in vitro findings (*Brancucci et al., 2017*) across a large malaria patient cohort. LPC is an immune effector molecule promoting macrophage polarization to M1 phenotype that induces the secretion of various cytokines such as IFN-γ and IL-1 family (i.e. IL-18) (*Huang et al., 1999*; *Qin et al., 2014*) through activation of the inflammasome in endothelial cells and peripheral blood mononuclear cells (PBMCs). Furthermore, LPC is the main component of the oxidized form of LDL (oxLDL) that induces inflammasome-mediated trained immunity in human monocytes (*Law et al., 2019*; *Amunugama et al., 2021*), resulting in increased responsiveness to LPS re-stimulation. Indeed, we demonstrate that LPC levels are positively associated with IFN-γ/IL-18 levels (Factor 5). These observations are in line with recent data from experimentally infected macaques and malaria patients, where LPC levels decreased in the acute phase compared to the pre-challenge and the chronic phase (*Cordy et al., 2019*). LPC is also a nutritional resource required by the parasite for replication (*Brancucci et al., 2017*) and hence scarcity is expected to promote reproduction, as gametocytes require less nutritional resource and, therefore, provide a better adaptation strategy.

Surprisingly, we also identified a link between *Pfsir2a* transcription, host inflammatory response, and parasite biomass (*Pf*HRP2). *Pf*Sir2a belongs to the evolutionarily conserved family of sirtuins that act as environmental sensors to regulate various cellular processes (*Li, 2013*; *Vasquez et al., 2017*; *Palacios et al., 2009*). In *P. falciparum*, PfSir2a and PfSir2b paralogues cooperate to regulate virulence gene transcription including *var* genes (*Tonkin et al., 2009*; *Duraisingh et al., 2005*). In vitro data have also demonstrated that increased PfSir2a levels are associated with reduced parasite replication (i.e. lower merozoite numbers) (*Mancio-Silva et al., 2013*). We hypothesize that the observed upregulation of *Pfsir2a* transcription in response to inflammation is part of an orchestrated stress response linking replication and antigenic variation (via *Pfsir2a*) to reproduction and transmission (via *ap2-g*), perhaps through a shared epigenetic control mechanism (*Coleman et al., 2014*). It is well known that host tolerance to malaria infection reduces with falling transmission (*Borrmann et al., 2011*; *Dollat et al., 2019*), as shown by the declining threshold of parasite biomass (*Pf*HRP2) required for clinical malaria. This suggests that parasites have more pronounced harmful consequences on the infected host (i.e. clinical symptoms) in low compared to high transmission settings, perhaps due to increasing host age (*Sorci et al., 2021*). Under this scenario, we propose that parasites experience increased within-host stress to which they respond through increased *ap2-g* transcription (to increase reproduction, hence transmission) and increased *Pfsir2a* transcription (to affect stress coping mechanisms at the expense of replication, hence the negative association with *Pf*HRP2) – as part of a self-preservation strategy in the face of imminent risk of host death.

In summary, we propose a model where the falling host immunity with declining transmission modifies the predominant host immune response, and consequently, the within-host environment (e.g. LPC availability, fever), resulting in increased investment in transmission (i.e. higher *ap2-g* transcription) and limiting replication (i.e. higher *Pfsir2a* transcription). Our findings provide critical information to accurately model parasite population dynamics. They suggest that parasite populations in elimination scenarios may increase their transmission potential. Understanding how malaria parasites adapt to their environment, for example by increasing investment in transmission stages at low endemicity, is highly relevant for public health. Not only would this affect the timelines for successful elimination,

but it would also form an important argument for the deployment of gametocytocidal drugs once transmission has been successfully reduced.

## Materials and methods

### Study design and participants

Ethical approval was granted by the Scientific Ethics Review Unit of the Kenya Medical Research Institute under the protocol; KEMRI/SERU/3149, and informed consent was obtained from the parents/guardian of the children. The study was conducted in Kilifi county which is a malaria-endemic region along the Kenyan coast. Over the last three decades, Kilifi has experienced changes in the pattern of malaria transmission and clinical presentation spectrum (*O'Meara et al., 2008*; *Mogeni et al., 2016*; *Njuguna et al., 2019*). The study included (i) children admitted with malaria at Kilifi county hospital (KCH) between 1994–2012 and recruited as part of the hospital admission surveillance system, (ii) children presenting with mild malaria at an outpatient clinic, and (iii) asymptomatic children which were part of a longitudinal malaria surveillance cohort which were sampled during annual cross-section bleed in 2007 and 2010. Clinical data, parasite isolates, and plasma samples collected from the children were used to conduct the study. The selection of sub-samples for quantifying inflammatory markers and lipids was informed by the availability of fever data and resources.

### Clinical definitions

Admission to malaria was defined as all hospitalized children with malaria parasitemia. The severe malaria syndromes: severe malarial anemia (SMA), impaired consciousness (IC), and respiratory distress (RD) were defined as hemoglobin <5 g/dl, Blantyre coma score (BCS) <5, and deep breathing, respectively (*Marsh et al., 1995*). Malaria admissions that did not present with either of the severe malaria syndromes were defined as moderate malaria. Mild malaria was defined as stable children presenting at the outpatient clinic with peripheral parasitemia, and asymptomatic as those with positive malaria (Giemsa smear) but without fever or any other sign(s) of illness. The combination of mild and moderate was referred to as uncomplicated.

### Controlled infection cohort

Malaria naïve volunteers were infected by either bite from five *P. falciparum* 3D7–infected mosquitoes (n=12) or by intravenous injection with approximately 2800 *P. falciparum* 3D7–infected erythrocytes (n=12); treatment with piperaquine was provided at a parasite density of 5000 /mL or on day eight following blood-stage exposure, respectively (*Alkema et al., 2022*).

### Parasite parameters

Thick and thin blood films were stained with Giemsa and examined for *P. falciparum* parasites according to standard methods. Data were presented as the number of infected RBCs per 500, 200, or 100 RBCs counted. This data was then used to calculate parasitemia per µl of blood using the formula described in '2096-OMS-GMP-SOP-09–20160222_v2.indd (who.int).' Briefly, parasites/µl=number of parasitized RBCs × number of RBCs per µl /number of RBCs counted or the number of parasites counted × number of WBCs per µl/number of WBCs counted. Where data on the actual number of RBCs or number of WBCs per µl of blood was not available, 5 million RBCs and 8000 WBCs per µl of blood were assumed.

### PfHRP2 ELISA

*P. falciparum* histidine-rich protein 2 (*Pf*HRP2) was quantified in the malaria acute plasma samples using ELISA as outlined. Nunc MaxiSorp flat-bottom 96-well plates (ThermoFisher Scientific) were coated with 100 µl/well of the primary/capture antibody [Mouse anti-*Pf*HRP2 monoclonal antibody (MPFM-55A; MyBioscience)] in 1 × phosphate buffered saline (PBS) at a titrated final concentration of 0.9 µg/ml (stock = 8.53 mg/ml; dilution = 1:10,000) and incubated overnight at 4 °C. On the following day, the plates were washed thrice with 1 × PBS/0.05% Tween-20 (Sigma-Aldrich) using a BioTek ELx405 Select washer (BioTek Instruments, USA) and blotted on absorbent paper to remove residual buffer. These plates were then blocked with 200 µl/well of 1 × PBS/3% Marvel skimmed milk (Premier Foods; Thame, Oxford) and incubated for 2 hr at room temperature (RT) on a shaker

at 500 rpm. The plates were then washed thrice as before. After the final wash, plasma samples and standards were then added at 100 ul/well and in duplicates. The samples and standards (*Pf*HRP2 Recombinant protein; MBS232321, MyBioscience) had been appropriately diluted in 1 × PBS/2% bovine serum albumin (BSA). The samples and standards were incubated for 2 hr at RT on a shaker at 500 rpm followed by three washes with 1 × PBS/0.05% Tween-20 and blotted dry as before. This was followed by the addition of a 100 μl/well of the secondary/detection antibody [Mouse anti-*Pf*HRP2 HRP-conjugated antibody (MPFG-55P; MyBioscience) diluted in 1 × PBS/2% BSA and at a final titrated concentration of 0.2 μg/ml (stock = 1 mg/ml; dilution = 1:5000)]. The plates were then incubated for 1 hr at RT on a shaker at 500 rpm, washed thrice as before, and dried on absorbent paper towels. o-Phenylenediamine dihydrochloride (OPD) (ThermoFisher Scientific) substrate was then added at 100 μl/well and incubated for 15 min for color development. The reaction was stopped with 50 μl/well of 2 M sulphuric acid ($H_2SO_4$) and optical densities (OD) read at 490 nm with a BioTek Synergy4 reader (BioTek Instruments, USA).

## Parasite transcript quantification using quantitative RT-PCR

RNA was obtained from TRIzol reagent (Invitrogen, catalog number 15596026) preserved *P. falciparum* positive venous blood samples obtained from the children recruited in the study. RNA was extracted by Chloroform method (*Bull et al., 2005*) and cDNA was synthesized using a Superscript III kit (Invitrogen, catalog number 18091050) following the manufacturer's protocol. Parasite gene transcription analysis was carried out through quantitative real-time PCR as described below.

Real-time PCR data was obtained as described (*Dondorp et al., 2005*; *Borrmann et al., 2011*; *Dollat et al., 2019*). Four primer pairs targeting DC8 (named dc8-1, dc8-2, dc8-3, dc8-4), one primer pair targeting DC13 (dc13), and two primer pairs targeting the majority of group A *var* genes (gpA1 and gpA2) were used in real-time PCR analysis as described (*Abdi et al., 2016*). We also used two primer pairs, b1, and c2, targeting group B and C *var* genes, respectively (*Rottmann et al., 2006*). Primer pairs targeting *Pfsir2a,* and *ap2-g* were also used (*Abdi et al., 2016*). Two housekeeping genes, Seryl tRNA synthetase and Fructose bisphosphate aldolase (*Salanti et al., 2003*; *Lavstsen et al., 2012*; *Abdi et al., 2016*) were used for relative quantification of the expressed *var* genes, *Pfsir2a,* and *ap2-g*. The PCR reaction and cycling conditions were carried out as described (*Lavstsen et al., 2012*; *Abdi et al., 2016*) using the Applied Biosystems 7500 Real-time PCR system. We set the cycle threshold (Ct) at 0.025. Controls with no template were included at the end of each batch of 22 samples per primer pair and the melt curves were analyzed for non-specific amplification. The *var* gene 'transcript quantity' was determined relative to the mean transcript of the two housekeeping genes, Sery tRNA synthetase and Fructose biphosphate aldolase as described (*Lavstsen et al., 2012*). For each test primer, the ΔCt was calculated relative to the average Ct of the two housekeeping genes which were then transformed to arbitrary transcript unit ($Tu_s$) using the formula ($Tu_s = 2^{(5-\Delta ct)}$) as described (*Lavstsen et al., 2012*). We assigned a zero $Tu_s$ value if a reaction did not result in detectable amplification after 40 cycles of amplification, that is, if the Ct value was undetermined.

## Measurement of cytokine levels in the plasma samples

The selection for this subset was primarily informed by the availability of fever data but the transmission period and clinical phenotype were also considered. However, there were more children with a fever data recorded in the post-decline period than in pre-decline and decline periods which biased the sampling toward the post-decline period. The plasma samples were analyzed using Procarta-Plex Human Cytokine & Chemokine Panel 1 A(34plex) [Invitrogen/ThermoFisher Scientific; catalog # EPX340-12167-901; Lot:188561049] following the manufacturer's instructions. The following 34 cytokines were measured: GM-CSF, IFN-α, IFN-γ, IL-1α, IL-1β, IL-1RA, IL-2, IL-4, IL-5, IL-6, IL-7, IL-8, IL-9, IL-10, IL-12 (p70), IL-13, IL-15, IL-17A, IL-18, IL-21, IL-22, IL-23, IL-27, IL-31, IP-10 (CXCL10), MCP-1 (CCL2), MIP-1α (CCL3), MIP-1β (CCL4), TNF-α, TNF-β, Eotaxin/CCL11, RANTES, GRO-a, and SDF-1a.

Briefly, 50 μl of magnetic beads mix was added into each plate well and the 96-well plate was securely placed on a hand-held magnetic plate washer for 2 min for the beads to settle. The liquid was then removed by carefully inverting the plate over a waste container while still on the magnet and lightly blotted on absorbent paper towels. The beads were then washed by adding 150 μl of 1 × wash buffer, left to settle for 2 min, and the liquid removed as before followed by blotting. This was followed by adding 25 μl of Universal Assay Buffer per well and then 25 μl of plasma samples and standards into

appropriate wells or 25 µl of Universal Assay Buffer in blank wells. The plate was covered and shaken on a plate shaker at 500 rpm for 30 min at room temperature followed by an overnight incubation at 4 °C. After the overnight incubation, the plate was shaken on a plate shaker at 500 rpm for 30 min at room temperature and the beads were then washed twice while on a magnetic plate holder as outlined above. The beads were then incubated in the dark with 25 µl of detection antibody mixture on a plate shaker at 500 rpm for 30 min at room temperature followed by two washes as before. A 50 µl of Streptavidin-Phycoerythrin (SAPE) solution was then added per well and similarly incubated for 30 min on a plate shaker at 500 rpm and at room temperature followed by two washes. After the final wash, the beads were resuspended in 120 µl of Reading Buffer per well, and incubated for 5 min on a plate shaker at 500 rpm before running on a MAGPIX reader running on MAGPIX xPOTENT 4.2 software (Luminex Corporation). The instrument was set to count 100 beads for each analyte. The analyte concentrations were calculated (via Milliplex Analyst v5.1 [VigeneTech]) from the median fluorescence intensity (MdFI) expressed in pg/mL using the standard curves of each cytokine.

## Lipidomics analysis

Patient temperature, *ap2-g* transcription level, and disease type were used to subset samples for lipidomics. This resulted in five groups from severe disease categories and matching mild cases (*Figure 4—figure supplement 1*). Plasma samples were preserved at –80 °C until extraction with the chloroform/methanol method. 25 µL of plasma was extracted with 1 mL of the extraction solvent chloroform/methanol/water (1:3:1 ratio), the tubes were rocked for 10 min at 4 °C and centrifuged for 3 min at 13,000 g. Supernatant was collected and stored at –80 °C in glass tubes until analysis.

Sample vials were placed in the autosampler tray in random order and kept at 5 °C. Separation was performed using a Dionex UltiMate 3000 RSLC system (Thermo Scientific, Hemel Hempstead) by injection of 10 µl sample onto a silica gel column (150 mm × 3 mm × 3 µm; HiChrom, Reading, UK) used in hydrophilic interaction chromatography (HILIC) mode held at 30°C (*Zheng et al., 2010*). Two solvents were used: solvent A [20% isopropyl alcohol (IPA) in acetonitrile] and solvent B [20% IPA in ammonium formate (20 mM)]. Elution was achieved using the following gradient at 0.3 ml/min: 0–1 min 8% B, 5 min 9% B, 10 min 20% B, 16 min 25% B, 23 min 35% B, and 26–40 min 8% B. Detection of lipids were performed in a Thermo Orbitrap Fusion mass spectrometer (Thermo Fisher Scientific Inc, Hemel Hempstead, UK) in polarity switching mode. The instrument was calibrated according to the manufacturer's specifications to give an rms mass error <1 ppm. The following electrospray ionization settings were used: source voltage, ±4.30 kV; capillary temp, 325 °C; sheath gas flow, 40 arbitrary units (AU); auxiliary gas flow, 5 AU; sweep gas flow, 1 AU. All LC-MS spectra were recorded in the range of 100–1200 at 120,000 resolutions (FWHM at m/z 500).

## Data preprocessing

The raw data were converted to mzML files using proteowizard (v 3.0.9706 (2016-5-12)). These files were then analyzed using R (v 4.2.1) libraries xcms (v 3.14.1) and mzmatch 2 (v 1.0–4) for peak picking, alignment, filtering, and annotations (*Chong and Xia, 2018*; *Scheltema et al., 2011*; *Smith et al., 2006*). Batch correction was applied as in (https://www.mdpi.com/2218-1989/10/6/241/htm), and the data was then checked using PCA calculated using the R function prcomp (see *Figure 4—figure supplement 1*). Data was then range normalized and logged transformed using MetaboanalystR (v3.1.0). The CHMI lipidomics data was analyzed the same way but did not require batch correction as the samples were run in one batch.

## Statistical analysis

Data analysis was performed using R (v4.2.1) except for the structural equation modeling which was done in Mplus8. We normalized non-normally distributed variables by log transformation. *qRT-PCR:* Zeros in qRT-PCR values were replaced by 0.001 (the value before log transformation as the smallest measured value is about 0.0017). The median transcript units from qRT-PCR were calculated as follows: DC8 median from four primer pairs used (DC8-1, DC8-2, DC8-3, and DC8-4) and group A median from three primer pairs (gpA1, gpA2, and dc13). Samples for which *ap2-g* or *pfsir2a* arbitrary transcript unit was greater or equal to 32 (that is the transcript quantity of the reference genes based on the formula (Tu$_s$ = 2$^{(5-\Delta ct)}$) *Zheng et al., 2010*) were deemed unreliable and excluded from the analysis that went into generating *Figures 2–4*. Comparison between the two groups was done

using a two-sided Wilcoxon test. All correlations were conducted using Spearman's rank correlation coefficient test. All forest plots were done using linear regressions adjusted for transmission period, *Pf*HRP2, and age of the patient (see figure legends) using R function lm. All multiple test corrections were done using Benjamini & Hochberg multiple tests (using R function p.adjust).

### Principal factor analysis

A measurement model (i.e. factor analytic model) was fitted to summarize the 34 analytes into fewer variables called factors. An exploratory factor analysis (EFA) was performed to explore the factor structure underlying the 34 analytes. Factors were retained based on the Kaiser's 'eigenvalue rule' of retaining eigenvalues larger than 1. In addition, we also considered the scree plot, parallel analysis, fit statistics, and interpretability of the model/factors. This analysis resulted in the cytokine data being reduced to five factors. This analysis was done using the R 'psych' library (v 2.1.9) available at https://CRAN.R-project.org/package=psych. The 34 analytes were individually linearly regressed to *ap2-g* or *Pfsir2* transcript levels with the transmission, *Pf*HRP2, and age correction (model: analyte ~transmission + *Pf*HRP2 +age). Then each factor was analyzed the same way.

### Lipidomics analysis

The preprocessed lipidomics data was tested using transmission period, *Pf*HRP2, and age-adjusted linear regression with any of the five factors. All m/z with a significant false discovery rate with any of the factors were then manually checked for peak quality and identified masses on mass and retention time (*Reis et al., 2013*). The remaining identified lipids were then checked for the relationship with *ap2-g* and *Pfsir2a* transcription levels (linear regression adjusted for transmission period, HRP, and age, see methods above). The CHMI lipidomics data was analyzed the same way but the peaks retained were those significantly different pre and post-treatment in either type of infection (student's t-tests corrected for multiple testing).

### Figures

All heatmaps were done using the R library pheatmap (v 1.0.12) available at https://CRAN.R-project.org/package=pheatmap, and all other plots using the R libraries ggplot2 (v 3.3.5) and ggpubr (v 0.4.0) available at https://CRAN.R-project.org/package=ggpubr.

## Acknowledgements

This work was supported by the Wolfson Merit Royal Society Award (to M.M.), Wellcome Trust Investigator Award 110166 (to F.A., L.S., J.L.S.F., and M.M.), and Wellcome Trust Center Award 104111 (to F F.A., L.S., J.L.S.F., and M.M.), Glasgow-Radboud collaborative grant (to M.A., T.B. and M.M.), European Research Council (ERC) Consolidator Grant (to T.B., ERC-CoG 864180 QUANTUM), Wellcome Award 209289/Z/17/Z (to A.A.) and a core Wellcome award to KEMRI-Wellcome Trust (203077/Z/16/Z). This paper was published with the permission of the director of Kenya Medical Research Institute.

## Additional information

### Funding

| Funder | Grant reference number | Author |
| --- | --- | --- |
| Wellcome Trust | 110166 | Fiona Achcar<br>Lauriane Sollelis<br>João Luiz Silva-Filho<br>Matthias Marti |
| Wellcome Trust | 104111 | Fiona Achcar<br>Lauriane Sollelis<br>João Luiz Silva-Filho<br>Matthias Marti |
| Royal Society | Wolfson Merit Award | Matthias Marti |

| Funder | Grant reference number | Author |
|---|---|---|
| European Research Council | | Teun Bousema |
| Wellcome Trust | 209289/Z/17/Z | Abdirahman I Abdi |

The funders had no role in study design, data collection and interpretation, or the decision to submit the work for publication. For the purpose of Open Access, the authors have applied a CC BY public copyright license to any Author Accepted Manuscript version arising from this submission.

## Author contributions
Abdirahman I Abdi, Conceptualization, Data curation, Formal analysis, Supervision, Funding acquisition, Investigation, Visualization, Methodology, Writing – original draft, Project administration; Fiona Achcar, Data curation, Formal analysis, Validation, Investigation, Visualization, Methodology, Writing - review and editing; Lauriane Sollelis, Supervision, Investigation, Methodology, Writing - review and editing; João Luiz Silva-Filho, Visualization, Writing - review and editing; Kioko Mwikali, Formal analysis, Investigation, Visualization, Methodology; Michelle Muthui, Shaban Mwangi, Hannah W Kimingi, Benedict Orindi, Amrita Chandrasekar, Formal analysis; Cheryl Andisi Kivisi, Formal analysis, Supervision; Manon Alkema, Formal analysis, Methodology, Writing - review and editing; Peter C Bull, Formal analysis, Investigation; Philip Bejon, Teun Bousema, Conceptualization, Writing - review and editing; Katarzyna Modrzynska, Conceptualization, Formal analysis, Investigation, Writing - review and editing; Matthias Marti, Conceptualization, Supervision, Funding acquisition, Visualization, Writing – original draft, Project administration

## Author ORCIDs
Fiona Achcar ⓘ http://orcid.org/0000-0001-8792-7615
João Luiz Silva-Filho ⓘ http://orcid.org/0000-0003-4762-2205
Shaban Mwangi ⓘ http://orcid.org/0000-0003-1961-3205
Manon Alkema ⓘ http://orcid.org/0000-0002-8906-0047
Teun Bousema ⓘ http://orcid.org/0000-0003-2666-094X
Matthias Marti ⓘ http://orcid.org/0000-0003-1040-9566

## Ethics
Ethical approval was granted by the Scientific Ethics Review Unit of the Kenya Medical Research Institute under the protocol; KEMRI/SERU/3149, and informed consent was obtained from the parents/guardian of the children.

## Decision letter and Author response
Decision letter https://doi.org/10.7554/eLife.85140.sa1
Author response https://doi.org/10.7554/eLife.85140.sa2

# Additional files

## Supplementary files
• Supplementary file 1. Associations between parasite parameters *ap2-g*, *Pfsir2-a* and *Pf*HRP2, and clinical parameters.

• Supplementary file 2. Associations between parasite parameters *ap2-g*, *Pfsir2-a* and *Pf*HRP2, host luminex markers, and lipidomics data.

• Supplementary file 3. Structural equation model (SEM). The model assumes that pre-existing host immunity affects the interaction between the host (i.e. altered within the host environment including inflammatory response. fever, nutritional resource availability) and parasite (i.e. altered investment in reproduction *vs* replication). Significant *p*-values are highlighted in bold and negative and positive estimates of associations are highlighted in blue and red, respectively.

• MDAR checklist

## Data availability

Raw data and script for all the analyses in this manuscript are available at https://doi.org/10.7910/DVN/BXXVRY. mzML mass spectrometry files are available at MetaboLights at https://www.ebi.ac.uk/metabolights/editor/study/MTBLS5130.

The following datasets were generated:

| Author(s) | Year | Dataset title | Dataset URL | Database and Identifier |
|---|---|---|---|---|
| Abdi AI, Achcar F, Sollelis L, Silva-Filho JL, Mwikali K, Muthui M, Mwangi S, Kimingi HW, Orindi B, Kivisi CA, Alkema M, Chandrasekar A, Bull PC, Bejon P, Modrzynska K, Bousema T, Marti M | 2023 | *Plasmodium falciparum* adapts its investment into replicationversustransmission according to the host environment | https://www.ebi.ac.uk/metabolights/editor/study/MTBLS5130 | EBI, MTBLS5130 |
| Abdi AI, Achcar F, Sollelis L, Silva-Filho JL, Mwikali K, Muthui M, Mwangi S, Kimingi HW, Orindi B, Kivisi CA, Alkema M, Chandrasekar A, Bull PC, Bejon P, Modrzynska K, Bousema T, Marti M | 2023 | Replication Data for: Plasmodium falciparum adapts investment into replication versus reproduction transmission according to host environment | https://doi.org/10.7910/DVN/BXXVRY | Harvard Dataverse, 10.7910/DVN/BXXVRY |

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
