## [Editor Report]

This important work provides insight into how *Plasmodium falciparum* optimises the balance between infection of the human host and investment in onward transmission to the mosquito. The results, based on appropriate and validated state-of-the-art methodologies, are convincing and have both theoretical and practical implications beyond a single subfield.

---

## [Decision Letter]

**Decision letter after peer review:**

Thank you for submitting your article "*Plasmodium falciparum* adapts its investment into replication versus transmission according to the host environment" for consideration by *eLife*. Your article has been reviewed by 3 peer reviewers, and the evaluation has been overseen by a Reviewing Editor and Dominique Soldati-Favre as the Senior Editor. The following individuals involved in the review of your submission have agreed to reveal their identity: Michael Delves (Reviewer #1); Robert William Moon (Reviewer #3).

1. Please respond to the Reviewers' comments concerning issues that require clarifications/revisions of the text, selected figures (e.g., Figure 2, 4), and figure legends.

2. Please include data that shows a lack of correlation between pfs16 and another gene that is not involved in gametocytes/sexual commitment.

3. Please revisit your statistical analyses as suggested.

*Reviewer #1 (Recommendations for the authors):*

I am not an expert in quantitative transcriptomics and lipidomics, however, the study appears to be well-designed, appropriately powered, and with appropriate controls and conclusions.

Line 35 – Please cite data from the most recent World Malaria Report.

Line 88 – Please explain how the 552 samples were selected. Was it at random?

Line 105 – Figure 2A and B both have three graphs. It would be helpful in the text to be clear when discussing the data, which graph in each panel is being referred to (i.e. left, middle, right).

Line 112 – "This association validates ap2-g as a proxy for both, stage conversion and gametocyte levels." This is likely the case, but the data presented here only shows ap2-g and Pfsir2a compared to pfs16. Do all genes show this positive correlation? The data would be strengthened by showing the (non)correlation between pfs16 and another gene not known to be involved in gametocytes/sexual commitment just to confirm that data observed is specific.

*Reviewer #2 (Recommendations for the authors):*

1. Most of the figures lack explanatory assays and are based on selected correlations. The authors should consider adding to the discussion not only a model but also ways to validate the model, to avoid misinterpretation of the results due to underlying causations.

2. Figure 1C, D lines 104-106: It is unclear why these sub-figures refer to outpatients as they describe the admissions. Please clarify.

3. Figure 2A lines 102-103: "For this purpose, we measured both total parasite biomass based on PfHRP2 levels and peripheral parasitemia based on parasite counts from blood smears". The authors should better explain how they linked PfHRP2 to the parasites' biomass. If the goal was to assess parasitemia in a different way, please show the ratio between units of PfHRP2 and parasitemia levels. The link between the two might not be linear.

4. Figure 2 and 4: The authors should revisit their statistical analyses throughout the manuscript. This is a supercritical element in such a field study.

Please provide an explanation for why you chose to use the Wilcoxon test for pair-wise comparisons. The Wilcoxson test is for non-parametric data; please demonstrate that the data does not distribute normally, for example, using the Shapiro-Wilk test. Furthermore, multiple comparisons for multifactorial data are usually done using multifactorial ANOVA analysis (more robust for false-positive errors).

5. Figure 2F: It is unclear, and again not explained, what are the described subgroups.

6. Figure 3: The figure legend does not note which correlation test was used. Please see major comment #4 and consider revising/explaining why you used the Wilcoxson test to analyze multifactorial data rather than a multifactorial ANOVA analysis. The authors should briefly explain the idea behind factorial analysis, as it will make the results clearer. For example, factor analysis is a derivative of the widely used PCA analysis for clustering multidimensional data.

7. Figure 4A: It is noted in the text that group F5 has a positive association with LPC.

However, according to the color-coding in the figure, the correlation appears to be closer to zero – again, a very odd interpretation.

8. Figure 4B: The explanation provided in the figure legend is unclear.

*Reviewer #3 (Recommendations for the authors):*

I am still not entirely clear on the description of the data from 4c-e, and they could be misinterpreted. The sentence "Indeed, LPC species showed a negative association with both ap2-g and Pfsir2a transcription levels (Figure 4C-E). The association was only significant in our data when inflammation is highest (and LPC level lowest), which is at low transmission (i.e., post decline)." Suggests there is a trend in both, with only post-decline significant. In fact, the predecline shows a non-significant positive correlation between AP2G and LPC so it's really quite different from the pattern seen post-decline. These sentences and discussion (L217) could be adjusted to make this clearer.

---

## [Author Response]

Reviewer #1 (Recommendations for the authors):I am not an expert in quantitative transcriptomics and lipidomics, however, the study appears to be well-designed, appropriately powered, and with appropriate controls and conclusions.Line 35 – Please cite data from the most recent World Malaria Report.

The numbers have been updated according to the WHO report from 2022.

Line 88 – Please explain how the 552 samples were selected. Was it at random?

The 552 samples were pragmatically selected so that the sample size is within available budget but the only consideration for inclusion was the clinical phenotype and transmission period so that the various clinical phenotypes and transmission period are adequately represented (Figure 1). This is noted in line 88f:

“552 of the admissions were pragmatically selected to ensure adequate sampling of the transmission periods and clinical phenotypes (Figure 1C).”

Line 105 – Figure 2A and B both have three graphs. It would be helpful in the text to be clear when discussing the data, which graph in each panel is being referred to (i.e. left, middle, right).

We have clarified this in the text.

Line 112 – "This association validates ap2-g as a proxy for both, stage conversion and gametocyte levels." This is likely the case, but the data presented here only shows ap2-g and Pfsir2a compared to pfs16. Do all genes show this positive correlation? The data would be strengthened by showing the (non)correlation between pfs16 and another gene not known to be involved in gametocytes/sexual commitment just to confirm that data observed is specific.

We would like to thank reviewer #1 for this important point. We are using the combination of two housekeeping genes (seryl tRNA synthetase, fructose biphosphate aldolase) to normalise all markers analysed (*ap2-g*, *Pfsir2a* and *Pfs16*, and *var* gene groups). We have now introduced an additional supplementary figure (Figure 3 —figure supplement 1) representing expression correlation of all three genes (*ap2-g*, *Pfsir2a* and *Pfs16*) with *var* gene groups. Indeed, *Pfs16* is only correlated with *var* groups B1 and C1 demonstrating that there is no systematic bias in the qRT-PCR data.

We have added the following text in the result section:

“To ensure that there is no systematic bias in the qRT-PCR data we also tested the association of *Pfs16* expression against *var* gene subgroups. *Pfs16* expression showed a significant positive correlation with group B (here termed gpb1) and C (gpc2) *var* genes, while no correlation was observed with the rest of the *var* subgroups (Figure 3 —figure supplement 1).”

Reviewer #2 (Recommendations for the authors):1. Most of the figures lack explanatory assays and are based on selected correlations. The authors should consider adding to the discussion not only a model but also ways to validate the model, to avoid misinterpretation of the results due to underlying causations.

That is a great suggestion and we have added a phrase to the discussion accordingly.

“We anticipate that the proposed model could be tested in controlled human malaria infections with malaria naïve and semi-immune participants as a proxy for low vs high transmission settings.”

2. Figure 1C, D lines 104-106: It is unclear why these sub-figures refer to outpatients as they describe the admissions. Please clarify.

Patient groups (admissions, outpatients and asymptomatics) and the clinical subgroups are all shown in Figure 1. The definition of these groups is included in the methods section under clinical definitions.

3. Figure 2A lines 102-103: "For this purpose, we measured both total parasite biomass based on PfHRP2 levels and peripheral parasitemia based on parasite counts from blood smears". The authors should better explain how they linked PfHRP2 to the parasites' biomass. If the goal was to assess parasitemia in a different way, please show the ratio between units of PfHRP2 and parasitemia levels. The link between the two might not be linear.

PfHRP2 has long been used as a more accurate marker for parasite biomass than peripheral parasitemia, both in *P. falciparum* (Dondorp *et al.*, PLoS Med, 2005; Marquart *et al.*, Mal J 2022) and in *P. vivax* (Barber *et al.* PLoS Path 2015, Silva Filho *et al. eLife* 2021) as PfHRP2 is secreted by sequestered and circulating parasites. As this reviewer notes the association between PfHRP2 and peripheral parasitemia is not necessarily linear (see references above).

We have added the original reference (Dondorp *et al.*, PLoS Med 2005) to the text.

4. Figure 2 and 4: The authors should revisit their statistical analyses throughout the manuscript. This is a supercritical element in such a field study.Please provide an explanation for why you chose to use the Wilcoxon test for pair-wise comparisons. The Wilcoxson test is for non-parametric data; please demonstrate that the data does not distribute normally, for example, using the Shapiro-Wilk test. Furthermore, multiple comparisons for multifactorial data are usually done using multifactorial ANOVA analysis (more robust for false-positive errors).

Generally, non-parametric tests are less powerful but more robust. We chose non-parametric tests since they are not affected by outliers as the variables do not perfectly follow normal distribution. Ttest produces similar result as Wilcoxon test. However, we used a parametric test (regression) where we needed to adjust for potential confounders such as transmission period, age etc (Figures 3 and 4).

5. Figure 2F: It is unclear, and again not explained, what are the described subgroups.

We included more details explaining the *var* subgroups in the figure legend.

6. Figure 3: The figure legend does not note which correlation test was used. Please see major comment #4 and consider revising/explaining why you used the Wilcoxson test to analyze multifactorial data rather than a multifactorial ANOVA analysis. The authors should briefly explain the idea behind factorial analysis, as it will make the results clearer. For example, factor analysis is a derivative of the widely used PCA analysis for clustering multidimensional data.

Both PCA and factor analysis are used to reduce high dimensional data, but factor analysis assumes that an interpretable latent variable (factor) explains the correlation between variables. For this reason, we attempted to assign the factors to biological function (line 148-152), and this is important in understanding the data presented in figure 3 and 4. We also used the data in structural equation modelling which relies on factor analysis.

7. Figure 4A: It is noted in the text that group F5 has a positive association with LPC.However, according to the color-coding in the figure, the correlation appears to be closer to zero – again, a very odd interpretation.

F5 association with the lipids has been done via a linear regression and associated statistic. Only one LysoPC species is significantly associated with F5 (LysoPC 18:2). This has been clarified in the manuscript and the associated statistics are now provided in supplementary file 2.

The following phrase has been edited accordingly:

“F5 showed significant positive association with one LPC species and negative associations with PC, respectively, while F4 showed significant negative association with three LPC species (Figure 4A, supplementary file 2).”

8. Figure 4B: The explanation provided in the figure legend is unclear.

We have added more detail to the annotation of this graph.

Reviewer #3 (Recommendations for the authors):I am still not entirely clear on the description of the data from 4c-e, and they could be misinterpreted. The sentence "Indeed, LPC species showed a negative association with both ap2-g and Pfsir2a transcription levels (Figure 4C-E). The association was only significant in our data when inflammation is highest (and LPC level lowest), which is at low transmission (i.e., post decline)." Suggests there is a trend in both, with only post-decline significant. In fact, the predecline shows a non-significant positive correlation between AP2G and LPC so it's really quite different from the pattern seen post-decline. These sentences and discussion (L217) could be adjusted to make this clearer.

We agree with this reviewer that the data in Figure 4C-E need clarification. Further analysis revealed that LPC levels are highest and vary little across patients at high transmission (pre decline). This lack of variation likely explains the lack of association with *ap2-g* transcription levels in the pre decline period. We have added a new panel in Figure 4D and text in the result section.

“In contrast, LPC levels are highest and vary little across patients at high transmission (pre decline).”

A new figure legend has been added accordingly:

“Distribution of *ap2-g* transcript levels (left) and LysoPC (16:0) intensity (right) across patients and stratified by transmission period. Data are corrected for age and *Pf*HRP2.”